# Renal Glucose Release after Unilateral Renal Denervation during a Hypoglycemic Clamp in Pigs with an Altered Hypothalamic Pituitary Adrenal Axis after Late-Gestational Dexamethasone Injection

**DOI:** 10.3390/ijms241612738

**Published:** 2023-08-13

**Authors:** Marius Nistor, Martin Schmidt, Carsten Klingner, Caroline Klingner, Matthias Schwab, Sabine Juliane Bischoff, Georg Matziolis, Guadalupe Leticia Rodríguez-González, René Schiffner

**Affiliations:** 1Orthopaedic Department, Jena University Hospital, Campus Eisenberg, 07607 Eisenberg, Germany; marius.nistor@uni-jena.de (M.N.);; 2Institute for Biochemistry II, Jena University Hospital, 07743 Jena, Germany; 3Department of Neurology, Jena University Hospital, 07747 Jena, Germanymatthias.schwab@med.uni-jena.de (M.S.); 4Institute for Laboratory Animals and Welfare, Jena University Hospital, 07743 Jena, Germany; 5Reproductive Biology, National Institute of Medical Science and Nutrition, 14080 Mexico City, Mexico; 6Emergency Department, Otto-von-Guericke University, 39120 Magdeburg, Germany; 7Emergency Department, Helios University Clinic Wuppertal, 42283 Wuppertal, Germany

**Keywords:** renal denervation, HPA axis, diabetes mellitus, hypoglycaemia, arterial hypertension, sympathetic nerves

## Abstract

Previously, we demonstrated in pigs that renal denervation halves glucose release during hypoglycaemia and that a prenatal dexamethasone injection caused increased ACTH and cortisol concentrations as markers of a heightened hypothalamic pituitary adrenal axis (HPAA) during hypoglycaemia. In this study, we investigated the influence of an altered HPAA on renal glucose release during hypoglycaemia. Pigs whose mothers had received two late-gestational dexamethasone injections were subjected to a 75 min hyperinsulinaemic–hypoglycaemic clamp (<3 mmol/L) after unilateral surgical denervation. Para-aminohippurate (PAH) clearance, inulin, sodium excretion and arterio–venous blood glucose difference were measured every fifteen minutes. The statistical analysis was performed with a Wilcoxon signed-rank test. PAH, inulin, the calculated glomerular filtration rate and plasma flow did not change through renal denervation. Urinary sodium excretion increased significantly (*p* = 0.019). Side-dependent renal net glucose release (SGN) decreased by 25 ± 23% (*p* = 0.004). At 25 percent, the SGN decrease was only half of that observed in non-HPAA-altered animals in our prior investigation. The current findings may suggest that specimens with an elevated HPAA undergo long-term adaptations to maintain glucose homeostasis. Nonetheless, the decrease in SGN warrants further investigations and potentially caution in performing renal denervation in certain patient groups, such as diabetics at risk of hypoglycaemia.

## 1. Introduction

In recent decades, there has been increasing evidence that deviations from the physiological prenatal glucocorticoid (GC) concentrations, whether of endo- or exogenous origin, such as malnutrition, stress or administration of synthetic glucocorticoids, exert an influence on foetal development [1]. Synthetic glucocorticoids, specifically bethamethasone and dexamethasone, have been widely used to accelerate lung maturation and to reduce complications associated with pre-term (24 to 34 weeks) birth; a 2017 Cochrane systematic review provided evidence for their effectiveness in reducing respiratory distress syndrome, neonatal death, intraventricular haemorrhage and 48 h postnatal systemic infection [2]. Although glucocorticoids have been used for this purpose for several decades, some specifics, such as treatment plans, dosages and questions regarding the beneficial effects of treating certain patient groups, remain controversial [3,4,5,6]. While the current risk-benefit profile of glucocorticoids in high-risk situations, such as imminent preterm birth, are clearly in favour of their application, there is evidence that antenatal glucocorticoids lead to an upregulation of the hypothalamic–pituitary–adrenal axis (HPAA) and low birth weight [7]. Several animal studies, as well as clinical and epidemiological studies in humans, have observed later-in-life pathologies and developmental issues, such as attention deficit hyperactivity disorder, behavioural issues and impaired cognitive abilities, as well as elevated cortisol concentrations (observed in rest, fasted state and during stress response), and an increased risk for the occurrence of cardiovascular disease and metabolic syndrome later in life, suggesting that increased antenatal glucocorticoid levels may have a significant epigenetic influence on the patient’s entire subsequent life [6,7,8]. However, in regard to the potential influence that long-term alterations of the HPAA have on certain treatment regimens or interventions, there still remains a large area of uncertainty, one of which concerns the relatively new treatment of renal denervation. In accordance with the general trend towards individualised medicine, it is crucial to investigate matters such as prenatal influences on the patient and their influence on the effectiveness (and potentially adverse effects) of later-in-life medical interventions. Autonomous sympathetic renal nerves exert an influence on blood pressure through modulation of the renin–angiotensin system, sodium reabsorption and changes in renal vascular tone, in addition to influencing general sympathetic activity via afferent fibres [9,10]. In recent years, renal denervation has been considered as a mechanistic treatment option for both treatment-resistant arterial hypertension and patients without antihypertensive drug treatment [11]. Further applications, such as in heart failure and treatment of arrythmias, are conceivable, but have not yet been investigated in large-scale experimental trials [9]. The initial hope for the procedure’s capability to significantly reduce arterial blood pressure was largely dampened by the SYMPLICITY HTN3 trial, among other contentious and unfavourable smaller studies [12]. Consequently, in 2018, the European Society of Cardiology (ESC)/European Society of Hypertension’s 2018 Task Force statement on treatment guidelines for arterial hypertension did not recommended renal denervation as a treatment option [13]. However, the SIMPLICITY HTN3 trial has since come under retrospective criticism for methodological and technical faults [10,14,15], and a number of subsequent studies—among them the SPYRAL HTN-OFF MED, SPYRAL HTN-ON MED and RADIANCE-HTN SOLO studies—have reported significant blood pressure reductions through renal denervation [16,17,18]. Consequently, in the newly published clinical consensus statement of the ESC council on hypertension and the European Association of percutaneous cardiovascular interventions of February 2023, renal denervation is recommended as an adjunct treatment option for patients with uncontrolled resistant hypertension despite lifestyle changes and drug treatment, and for those unable to tolerate long-term antihypertensive pharmacological treatment [11].

Since hypertension and diabetes mellitus are two closely related, often co-occurring diseases that exert reciprocal influences on their respective progressions and potentially share, as mentioned previously, at least in some cases, an etiological mechanism (prenatal), the influence of renal denervation on glucose homeostasis and response has to be examined, especially if the procedure becomes more common in the future and could potentially be applied in diabetic patients [19,20,21].

The liver and the kidney are the only organs in the human body capable of significant gluconeogenesis and subsequent glucose release into the bloodstream, as all other tissues lack the enzyme glucose-6-phosphatase necessary for glucose release. In the past, it was believed that the liver played a disproportionately larger role in maintaining glucose homeostasis, but, in recent decades, increasing evidence has shown that the kidneys also play a crucial role [22]. Renal glucose release and homeostasis are modulated by substrate availability (lactate, amino acids, etc.), glucose transporters (GLUT1, SGLT1 and SGLT2), hormones (stress hormones such as stress hormones, insulin and catecholamines), sympathetic activation, and hepato–renal reciprocity, although the proportions and exact mechanisms of renal glucose homeostasis are not fully understood [23,24]. 

Hypoglycaemia is one of the most common and dangerous side-effects of anti-diabetic treatment. In previous experiments, we were able to demonstrate that exposure to late-term GC causes elevated concentrations of ACTH and cortisol both at baseline and during hypoglycaemia in pigs as compared to control animals [25]. Moreover, in a different study, we showed that in pigs with unilateral renal denervation, the renal glucose release of the ablated kidney during conditions of severe hypoglycaemia was significantly diminished compared to that of the non-ablated organ. The two main stress systems of the human body, the HPAA and the autonomic nervous system, have long been thought to be closely interconnected, such that a change in one system may be expected to result in further changes in the other [26,27,28,29]; furthermore, they are known to communicate closely and interrelatedly in energy homeostasis [30,31]. Due to the frequent comorbidity of hypertension and diabetes mellitus type 2, it is crucial to investigate the currently unclear risk profile that renal denervation constitutes in regard to the body’s capacity for counter-regulation during hypoglycaemic episodes, a common side effect of antidiabetic treatments. 

Therefore, we hypothesized that a late-gestational glucocorticoid exposure would alter the magnitude of glucose release from a nerve-ablated kidney during a hypoglycaemic episode in an animal model of unilateral renal denervation in pigs.

## 2. Results

To elucidate whether a prenatal maternal glucocorticoid injection exerts an influence on the extent of renal gluconeogenesis during hypoglycaemia in the offspring after renal ablation, we subjected nine animals to a hyperinsulinaemic–hypoglycaemic clamp after open-access, unilateral surgical removal of the renal plexus. 

Compared to baseline, the hypoglycaemic procedure caused an increase in heart rate (mean increase 88 bpm), an increase in blood pressure (mean systolic increase 20 mmHg), and a decrease in body temperature (mean decrease of 1.8 °C) as measured at the end of the experimental procedure (Table 1). No significant differences in side-dependent urine volume were observed over the course of the experimental procedure (Figure 1).

Hypoglycaemia was deemed to be established at blood glucose levels below 3 mmol/L and maintained for 75 min (Figure 2). Mean blood glucose decreased from 7.7 ± 0.7 mmol/L to 0.69 ± 0.52 mmol/L. Since there were slight variations in the time necessary to establish hypoglycaemia, all subsequent measurements refer to the first time a blood glucose level below 3 mmol/L was measured.

PAH concentrations under established hypoglycaemia did not differ significantly between ablated and non-ablated kidneys in all consecutive measurements made every fifteen minutes after initial establishment of hypoglycaemia (Figure 3A). Plasma concentrations of PAH were maintained at about 15 µg/mL (Figure 3B). As a result, there were no significant differences between ablated and non-ablated kidneys in regard to side-dependent renal plasma flow (SRP) (Figure 3C).

Similarly, no significant differences were observed in regard to inulin (Figure 4A) and glomerular filtration rate (GFR) (Figure 4B). Measurements of SRP, inulin and GFR surgical intervention did not significantly alter physiological renal function.

Compared to the non-ablated side, renal denervation caused a significant decrease in side-dependent renal net glucose release (SGN) as computed by a Wilcoxon signed-rank test (*p* = 0.004), both in regard to absolute glucose release at all time points and relative to the non-ablated side, with an average decrease in SGN of 25 ± 23% (Figure 5). Over time, no significant interactions were observed between the respective time points of measurement and kidney side (ablated/non-ablated), resulting in a nearly constant decrease in SGN over the course of the experimental procedure.

Renal denervation significantly increased side-dependent urinary sodium concentrations at all time points of the experimental procedure (Figure 6), as confirmed by a comparison of ablated and non-ablated kidney with a Wilcoxon signed-rank test (*p* < 0.01). 

## 3. Discussion

In a pig animal model that had received a late-gestational dexamethasone injection, a postnatal hypoglycaemic clamp procedure after unilateral renal denervation showed a marked decrease in side-dependent renal net glucose release (SGN) in the ablated kidney as compared to the control (non-ablated kidney). The average decrease of 25% is half of what we found in our previous investigation using the same experimental model of pigs that did not receive a prenatal glucocorticoid (GC) dose [32]. As described before, we administered two dexamethasone injections with a twelve-hour interval on the 99th and 100th day of gestation at a dose of 0.06 mg/kg body weight. This is a lower dose than the corresponding human dosage of 6 mg dexamethasone (equivalent to 0.075 mg/kg body weight) in an 80 kg patient, which we nonetheless showed to cause a measurable change in the hypothalamic pituitary adrenal axis (HPAA) through later-in-life measurement of baseline hormone levels in our established experimental setup [25]. Animals undergoing the same prenatal treatment of glucocorticoid injections showed a pronounced elevation of baseline ACTH (140 ± 13 pg/mL vs. control: 42 ± 7 pg/mL) and cortisol (483 ± 30 nmol/L vs. control: 258 ± 27 nmol/L) [25]. One limitation of this study is that we did not measure the hormonal response of the animals once more, either at baseline or during the hypoglycaemic episode. However, we did not expect a significant change in hormonal response solely from the surgical procedure, as we did not manipulate the nerval fibres of the celiac ganglion or operate on the adrenal glands. Between the study population of our prior investigation in non-HPAA-altered animals and this current study exist only slight differences pertaining to body weight and age of the piglets. In our prior study, the piglets had an average body weight of 38 ± 7 kg [32] vs. 27 ± 6 kg in this study, and were 85 ± 10 days [32] at the age of the experimental procedure vs. 78 ± 7 days in this current study. The age difference can be assumed to be negligible due to the close proximity in respect to the developmental stage of the piglets. The difference in body weight can be explained in the context of the typical—and expected—changes effected through an alteration of the HPAA, and corresponds to those observed in humans as well. The experimental protocol of our prior investigation [32] was adopted to a large extent in this current study. The body temperature of the animals, who received a large abdominal laparotomy, decreased only moderately; from 37.5 ± 0.5 to 36.1 ± 0.7 °C in our prior investigation [32] and from 37.6 ± 0.6 °C to 35.8 ± 0.8 °C in this current study.

The relatively large standard deviation of this study is due to an increase in SRP and urine volume—with a concomitant increase in GFR and, due to the mathematical model employed, SGN—in two animals at consecutive time points of the experimental procedure, for which, unfortunately, we can provide no conclusive explanation. These two outliers, however, somewhat blunt the overall statistical result. 

As in our previous study, we used PAH clearance to calculate the effective plasma flow, which, at 15 µg/mL, was significantly below the saturating concentration of approximately 100 µg/mL. Equally, we observed no significant alterations of GFR and side-dependent renal plasma flow (SRP) of the ablated kidneys that would explain the decrease in SGN through changes in haemodynamics caused by the denervation procedure. This current experiment is in accordance with both our prior investigations and other descriptions of post-procedural renal blood flow and GFR, as well as SRP during hyperperfusion, which was necessary to maintain a stable blood pressure [32,33,34,35,36]. Compared to our investigations in the same experimental model of unilateral denervation in animals without prenatal GC administration, heart rate was significantly elevated both at baseline (124 ± 7 vs. 111 ± 7 in [32]) and during the hypoglycaemia (185 ± 8 vs. 124 ± 15 in [32]). 

As described previously, an altered HPAA can be induced by a variety of endogenous and exogenous prenatal conditions, and is suspected to exert significant influence on later-in-life physiological and psychological responses and susceptibility to certain pathologies [6,7,8,37,38,39,40,41,42,43]. Although we did not measure heart rate variability as an indicator of a concomitant heightened state of the autonomic nervous system (ANS) to the recorded HPAA elevation, the observed baseline vital parameters allow an indirect conclusion on the close interrelatedness of the later-in-life ANS to prenatal HPAA alteration, which furthermore has previously been described in the literature [27,44,45]. 

In this current study, we were able to show that prenatal GC treatment alters the SGN capacity of a denervated kidney during hypoglycaemia. It is noteworthy that our results comprise relative measurements in comparison to the non-denervated organ; the experimental setup is not capable to factor in side-dependent glucose consumption of renal cells or glucose reabsorption, though these factors should be negligible insofar as no significant changes can be expected between the surgically and non-surgically prepared organ. Surprisingly, the extent of the decrease in SGN in the denervated kidney was only half of that observed in our previous study in non-stress-axis-altered specimens. Nonetheless, HPAA-altered animals did not exhibit a significantly heightened blood sugar at baseline as compared to the control non-glucocorticoid-treated animals in our prior investigations [32]. One possible explanation for this less pronounced decrease in SGN is that, in order to maintain glucose homeostasis, the extent of sympathetically triggered renal gluconeogenesis is blunted in favour of other regulatory mechanisms, either intrarenally or correlated with adaptive processes in the liver, the other main homeostatic agent of the glucose metabolism. In the past, it was believed that the liver played a disproportionately larger role in maintaining glucose homeostasis, but in recent decades, increasing evidence has shown that the kidneys also play a crucial role. The proportion of renal gluconeogenesis appears to correspond to the liver’s current needs and demands and vice versa (hepatorenal glucose reciprocity), such that in a prolonged fasting post-absorptive state, as the liver’s glycogen storage is depleted, the proportion of renal gluconeogenesis in the body’s glucose supply rises; and likewise, in the postprandial state, renal gluconeogenesis doubles to account for up to 60% of the total gluconeogenesis, likely allowing for the replenishment of the liver’s glycogen reserves [22,23]. One conceivable mechanism of shifting the relative proportions of gluconeogenesis in favour of hepatic gluconeogenesis might be due to the effects of glucocorticoids themselves. Glucocorticoids are known to trigger transcription factors for gluconeogenic pathways such as pyruvate carboxy kinase, glucose-6-phosphatase and fructose-1,6-bisphosphatase [46]. Dexamethasone itself has recently been shown, alongside prolonged states of fasting, to induce the gluconeogenic genetic pathways through promotion of krüppel-like-factor 9 [47]. 

Hepatorenal reciprocity is a concept that is not yet fully understood, but, in inverted situations—where the liver’s capacity to ensure glucose homeostasis and respond to disruptions such as hypoglycaemia is diminished—a reciprocal shift has been described in which renal gluconeogenesis gains importance in homeostatic maintenance [23]. Under the influence of a persistently elevated sympathetic tone, it is conceivable that an inverse shift in favour of hepatic regulatory mechanisms occurs. However, our experimental model was not designed to detect these potential adaptive processes.

While blood homeostasis in pigs and humans shares many similarities, a direct transfer of our experimental results is limited [48]. Firstly, the employed surgical procedure does not correlate to that performed in humans [11,49,50], as we used an open abdomino-retroperitoneal access and performed a complete removal of the adventitial layer of the renal arteries. The advantage of this surgical procedure is the certainty of a complete removal of the nerval plexus, whereas the results of earlier studies on the effects of renal denervations in humans—among them the SYMPLICITY HTN3 trial—were extenuated by an uncertainty regarding the completeness and effectiveness of nerval ablation [10,11,14,15]. To further validate the successful execution of the intervention, we utilised sodium excretion as a functional parameter of the evidence of the denervation procedure, as the severing of the renal plexus leads to a heightened sodium excretion (with a concomitantly elevated water excretion and subsequent drop in blood pressure, which we counteracted as described in the “Methods” section). A further limitation of this study is the unilateral procedure of renal denervation, which is not performed in humans where ablation is performed bilaterally. A unilateral experimental model allows an intraindividual comparison of the respective organs, thereby excluding interindividual differences. Nonetheless, it is conceivable that the randomisation process used to determine the site of ablation obfuscated the magnitude of the decrease in the SGN, since Imnadze et al., reported that there are side-specific differences in the extent of sympathetic innervation, favouring the right kidney over the left [51]. 

Moreover, since this investigation was primarily focused on the effects of HPAA alteration on the renal capacity to respond to hypoglycaemia and the physiological mechanisms through which this is mediated, our experimental model does not adequately reflect the pathophysiological correlate that one might expect to find in humans undergoing an ablation procedure. Hypertension and diabetes exhibit reciprocal disease progressions and pathophysiological influences, and often co-occur with chronic kidney disease, either as a complication of the aforementioned pathologies or as a result of unrelated causes. Each of these conditions alter renal function, cause adaptions and remodelling processes and decrease renal capacity for gluconeogenesis [52,53,54,55,56,57]. A potential further decrease in the renal capacity to counter hypoglycaemic states through gluconeogenesis may therefore be assumed in the context of human patients having undergone renal denervation, who, due to chronic disease, already have a reduced kidney function or sustained structural organ damage. None of the aforementioned pathologies were induced externally in the animals used in this study, and the time of the experiments in relation to the pigs’ age (78 ± 7 days) should not have been sufficient to cause chronic pathologies, even if these could be hypothesized to occur in older specimens as a result of a chronically elevated HPA axis.

Many patients at risk for hypoglycaemia, namely diabetics with antidiabetic drug treatment, are frequently treated with a multi-drug regimen, and interactions and confounding factors of commonly prescribed antidiabetic agents should be considered when evaluating the renal capacity for gluconeogenesis. At least for the relatively new and very promising class of SGLT-2-inhibitors, there is some evidence of an intrinsic upregulation of renal gluconeogenesis through mechanisms not yet fully understood [58,59]. Solis-Herrera et al. described an increase in endogenous glucose production in patients who had undergone kidney transplantation (equivalent to renal denervation) following administration of dapagliflozine; however, an interference by a vestigial sympathetic activity of the remaining native kidney could not be ruled out [60]. Since insulin itself decreases renal gluconeogenesis (among other mechanisms via a decrease in gluconeogenic substrates and diminished expression of PCK1 and G6P mRNA), it might be hypothesized that hypoglycaemia of origins other than insulin mediation (sulfonylureas, chronic liver disease, etc.) exhibits a different magnitude of renal SGN response. 

All of our experiments were conducted during daytime at approximately the same time in the early afternoon. The HPAA, and cortisol as its effector hormone, underlies a diurnal circadian rhythm [61,62]. If there is a relationship between the HPAA and the ANS, it can be assumed that there is a significant degree of variability in the amount of sympathetically influenced renal counterregulatory measures to hypoglycaemia at different times of the day. Furthermore, the magnitude of the HPAA response to external stressors depends on whether it occurs during the rising or falling phase of the ultradian pulse [62]. Since severe hypoglycaemic episodes frequently occur at night, a diminished renal SGN response is conceivable, which would be of particular relevance since severe (primarily nocturnal) hypoglycaemic episodes are suspected as a cause of death in up to 4–10% of diabetic patients treated with antidiabetics [63].

Our experiments were conducted following a twenty-four-hour food withdrawal. It is known that prolonged fasting increases the share of renal gluconeogenesis in total endogenous glucose production, so that after forty-eight hours, it amounts to approximately fifty percent of total glucose homeostasis [23,24]. The baseline parameters of this experimental setup may therefore be limited by a rather uncommon equivalent situation in humans, where hypoglycaemic episodes (through iatrogenic insulin treatment) rarely occur in such a prolonged fasting state. 

Ultimately, further investigations into a blunted response, and therefore the potentially deleterious side effect of prenatal glucocorticoid treatment, of renal gluconeogenesis should be conducted in experimental models that mimic the specific pathophysiology most likely to be found in the affected patient group, primarily a disease model of simultaneous hypertension and diabetes. 

## 4. Materials and Methods

### 4.1. Experimental Animals, Surgical Procedures

All procedures were approved by the Saxony animal welfare committee (Leipzig; permission number: TVV 07/13). All animals were of the German landrace, and the offspring were reared in a conventional agricultural holding facility. During their upbringing, the pigs received regular fodder. 

Nine time-dated pregnant sows were randomly assigned out of a group we have previously described in detail [25]. Both the piglets utilised in our prior study investigating the effects of renal denervation on renal glucose release without an altered HPA axis [32] and those of this current study were chosen from this pool of offspring [25]. The pregnant pigs received an intramuscular injection of 60 μg/kg body weight dexamethasone (Dexamethason Injektionslösung ad us. Vet^®^, CP-Pharma, Burgdorf, Germany) on the 99th and 100th day of gestation (dGA, term = 154–156 days) with 6–10 mL stock solution. Surgery was performed on pigs weighing between 19.5 and 39.8 kg in strict accordance to local standards and the “*Guide for the Care and Use of Laboratory Animals*” [64] (Table 1). 

The experimental procedure was preceded by a 24 h food withdrawal, during which the animals had ad libitum access to water. Surgery was performed in supine position under sterile conditions and general anaesthesia. The room temperature of the operating room was 24 °C. Initial anaesthesia was achieved via intramuscular injection of 15 mg/kg ketamine hydrochloride (Ketavet^®^, 100 mg/mL, Pharmacia Upjohn, Erlangen, Germany) and 0.2 mg/kg midazolam hydrochloride (Midazolam-ratiopharm^®^, Ulm, Germany). Subsequently, 0.2–0.3 mg/kg propofol (Disoprivan^®^, AstraZeneca, Wedel, Germany) was administered through a venous catheter in the ear vein (Vasocan^®^, Braun Melsungen, Melsungen, Germany). Intubation was performed orotracheally (Trachealtubus, Rüsch, Kernen, Germany). Maintenance of anaesthesia was ensured via continuous inhalation of 1.5% isoflurane (Isofluran^®^, DeltaSelect, Dreieich, Germany) and O_2_. An intravenous application of 0.003 mg/kg fentanyl per hour (0.05 mg/mL Fentanyl, Janssen) ensured analgesia. Muscular relaxation was induced by administration of up to 0.1 mg/kg pancuronium (Pancuronium-Actavis^®^, Actavis, München, Germany). By administration of eye drops (Corneregel^®^, Bausch&Lomb, Berlin, Germany), the cornea were kept moistened. A stable blood pressure was maintained via intravenous administration of isotonic saline (Isotonische Kochsalzlösung^®^, Fresenius, Bad Homburg, Germany) during anaesthesia. The saline infusions were preheated to a temperature of 30 °C. For blood sampling and blood pressure measurement, vascular catheters were inserted into the carotid artery and the jugular vein (Arteriofix^®^, Braun, Melsungen, Germany) for intraoperative administration of drugs and fluid infusions. Surgical access was gained via open abdomino-retroperitoneal entrance. To achieve this, a midline laparotomy and retroperitoneum opening was performed. Size 6 Charier catheters (Actreen^®^ Glys Cath, Braun, Melsungen, Germany) were inserted in both ureters for urine sampling. For local blood sampling, the renal veins were bilaterally instrumented with vascular catheters (Certofix^®^ Trio, Braun, Melsungen, Germany) and held in place with Liquiband^®^ Flow Control (Advanced Medical Solution, Devon, UK). The side of the unilateral renal denervation had been determined beforehand via a randomization list. During the surgical preparation, utilising an open abdominal access, the neuroplexus of the arteria renalis was severed under visual control on the previously determined side. Fibres of the celiac ganglion which primarily innervate the adrenal gland were not manipulated. A subsequent direct control of the success of the severing of the neuroplexus was not performed; however, the measured altered sodium excretion of the denervated kidney allows the conclusion that the renal denervation was successful in each individual case. 

After the experiments had been concluded, the animals were euthanized painlessly via intravenous administration of 60 mg/kg pentobarbital (Narcoren^®^, Merial, Hallbergmoos, Germany).

### 4.2. Induced Polyuria and Maintenance of Blood Pressure

Hyperperfusion of the kidneys was incumbent to avoid the need for exogenous catecholamine administration for the maintenance of a stable blood pressure and obfuscation of the investigated physiological mechanisms. High volume infusions of sterile isotonic saline were used to induce a polyuria of more than 100 mL/hour/kidney; infusion rates varied between 1 and 2 litres/hour and were adjusted to invasive blood pressure measurements and urine production. In cases of pronounced decrease in urine volume, furosemide (Furosemid-ratiopharm^®^, 40 mg/mL, Ratiopharm, Ulm, Germany) was administered in doses ranging from 40 to 120 mg per animal.

### 4.3. Hypoglycemic Clamp

Hypoglycaemia was initially induced by a bolus of 15 IU human regular insulin (Actrapid^®^, Penfill^®^, 100 IU/mL, Novo Nordisk Pharma, Mainz, Germany) through the central venous catheter. Arterial blood glucose levels were monitored with a blood glucose meter (Contour^®^, Bayer AG, Leverkusen, Germany) every 7.5 min. If necessary, additional dosages were administered. Total individual insulin dosages varied within a range of 20–100 IU of insulin per experimental procedure. 

### 4.4. Determination of Side-Dependent Glomerular Filtration Rate, Renal Plasma Flow and Gluconeogenesis

Inulin and para-aminohippurate (PAH) values were adjusted near equilibrium through a two-step protocol. This required an initial bolus of 3.1 g PAH (Sigma-Aldrich, Taufkirchen, Germany) and 0.55 g inulin (Sigma-Aldrich, Taufkirchen, Germany), followed by a continuous infusion of consisting of 4.3 g PAH and 1.1 g inulin at a rate of 250 mL/hour until the end of the experimental procedure. More details about this protocol are provided in our previous study, in which we utilized the same procedure [32]. 

The side-dependent renal plasma flow (SRP) was determined separately for each kidney, using the equation SRP [mL/min] = C_UrinePAH_ [mg/L] × V_UrineVolumeOverTime_ [mL/min]/C_PlasmaPAH_ [mg/L]/0.9 [65,66]. Urine specimens for the analysis of C_UrinePAH_ were collected via catheters from both ureters. For the analysis of C_PlasmaPAH_, blood specimens were drawn from the carotid artery. Urine and blood specimens were collected every 15 min. V_UrineVolumeOverTime_ was measured via the urine collected in the ureter catheters. The urine of each kidney was collected in intervals of 15 min and its volume was measured. The glomerular filtration rate (GFR) was determined separately for each kidney, using the equation C_UrineInulin_ [mg/L] × V_UrineVolumeOverTime_ [mL/min]/C_PlasmaInulin_ [mg/L] [67]. Urine specimens for the analysis of C_UrineInulin_ were collected via catheters from both ureters. For the analysis of C_PlasmaInulin_, blood specimens were drawn from the carotid artery. Urine from each kidney and blood specimens were collected every 15 min. V_UrineVolumeOverTime_ was determined using the ureter catheters. The side-dependent renal glucose release (SGN) was calculated for each kidney separately, using the equation SGN [mmol/min] = SRP [L/min] × (C_VenousGlucose_ [mmol/L] − C_ArterialGlucose_ [mmol/L]). C_VenousGlucose_ was measured in blood specimens drawn from catheters in both renal veins. Blood specimens for the analysis of C_ArterialGlucose_ were drawn from a catheter in the carotid artery.

### 4.5. Pre-Analytical Methods

Similar to our previous study, all plasma and urine samples were aliquoted after collection and stored until analysis. For PAH measurements, samples were thawed in a water bath, and subsequently vortexed and centrifuged in a microcentrifuge at room temperature. The cleared supernatants were used for further analyses. Reagents were of analytical grade (purchased from Roth, Karlsruhe, Germany).

### 4.6. Quantitation of Inulin

We adapted the quantitation method described by Roe et al. [68] to enable the use of a microplate reader. After sample centrifugation for 10 min, the deproteinized supernatants were used undiluted (plasma) or routinely diluted 1:10 (urine) on the animals and the sites of sample taking. For a more detailed description of the quantitation method, we refer to the in-depth explanations in our previous study [32]. 

### 4.7. Quantitation of PAH

The established method described by Agarwal et al. was used [69]. Briefly, this method uses a microplate assay based on the reaction of p-dimethylaminocinnamaldehyde (Sigma-Aldrich, Taufkirchen, Germany) with PAH. The results thus obtained are in good agreement with HPLC-based methods. For a more detailed description of the quantitation method, we refer to the in-depth explanations in our previous study [32].

### 4.8. Quantitation of Sodium Excretion

A routine clinical procedure was used to measure sodium concentrations in the urine samples.

### 4.9. Statistical Analyses

Data were tested for normal distribution with the Shapiro–Wilk test. At some time-points, data did not follow a normal distribution. Therefore, to investigate the differences in renal gluconeogenesis between the ablated and non-ablated kidneys during hypoglycaemia, a Wilcoxon signed-rank test was employed as part of our analytical approach. The purpose of this test was to compare the glucose synthesis capabilities of the two kidney sides within each animal.

To begin with, we computed the mean values for each animal and kidney by averaging the results across all measurement time points. This yielded a single representative value for both the ablated and non-ablated sides of the kidneys for each animal. By summarizing the data in this manner, we aimed to obtain a comprehensive assessment of gluconeogenic activity on each side.

Next, the computed mean values were subjected to a Wilcoxon signed-rank test to quantify the disparities in renal gluconeogenesis between the ablated and non-ablated sides. The paired design of the Wilcoxon signed-rank test allowed us to control for individual variations and focus specifically on the differences between the kidney sides within the same animal. The significance level was set at α = 0.05. All statistical analyses were carried out using R 4.3.1.

The raw data of side-dependent renal glucose release after unilateral renal denervation can be found in the Appendix A. 

## 5. Conclusions

In conclusion, we were able to demonstrate that a prenatal glucocorticoid administration and concomitant alteration of the hypothalamic pituitary adrenal axis (HPAA) alter the side-dependent glucose release (SGN) in a pig model of unilateral renal denervation. Since the observed decrease in SGN through ablation is on average 25 percent, which is half of that observed in animals without an altered HPAA, the results of this study warrant further investigations into potential adaptive processes in the renal glucose metabolism exerted through chronically elevated states of the HPAA and the autonomous nervous system (ANS). Even if the decrease in SGN is less than that observed in our prior investigation, this current experimental model may be closer to the pathophysiological correlate in humans, as an elevated HPAA is thought to contribute to the development of treatment-resistant hypertension and diabetes. The former represents the main indication of a renal denervation procedure in humans. The latter constitutes, as an adverse effect of common treatment regimens with insulin, one of the main causes of episodes of hypoglycaemia. If our results are confirmed in future investigations, caution may be advised regarding the inclusion of specific patient groups for renal denervation, such as those with specific patient histories (known premature delivery and maternal glucocorticoid treatment), certain conditions such as hypercortisolism, as well certain antidiabetic regimens such as intensified conventional therapy, especially in the elderly, those with hypoglycaemia unawareness, or diminishing cognitive abilities.

## Figures and Tables

**Figure 1 ijms-24-12738-f001:**
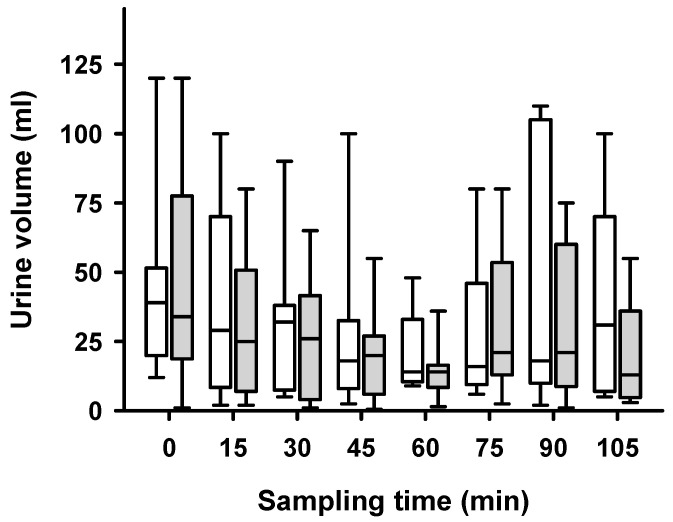
Side-dependent urine volume over time after unilateral ablation of renal nerves. Measurement of side-dependent urine volume (SRP) (mL) during hypoglycaemia showed no differences between non-ablated (white boxes) and ablated (grey boxes) kidneys. All measurements of SRP were performed during conditions of a high-volume infusion which induced hyperperfusion of the kidneys. Samples were taken every 15 min. Data are presented as box plots, showing medians, 25/75th percentiles (boxes), and 10/90th percentiles (whiskers), *n* = 9, *p* > 0.05.

**Figure 2 ijms-24-12738-f002:**
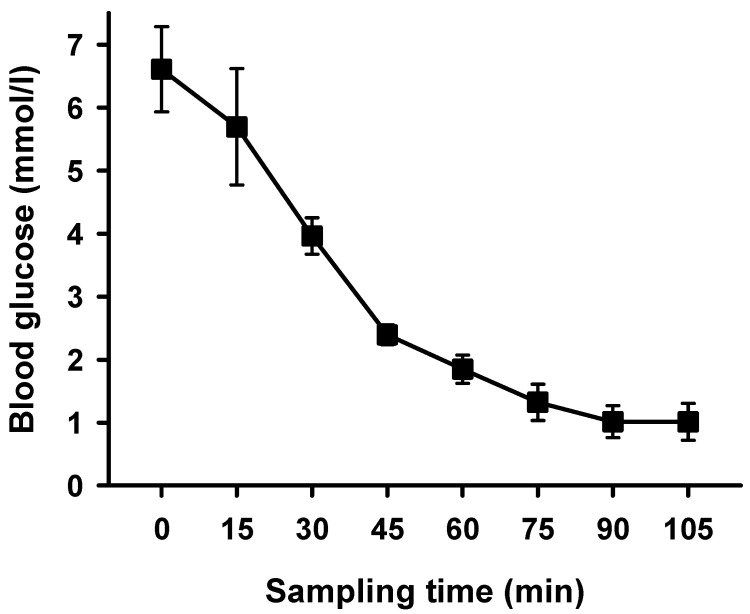
Mean blood glucose during hypoglycaemic clamp after unilateral ablation of renal nerves. Measurements of mean blood glucose [mmol/L] performed every 15 min, beginning with first administration of insulin; means ± SEM, *n* = 9.

**Figure 3 ijms-24-12738-f003:**
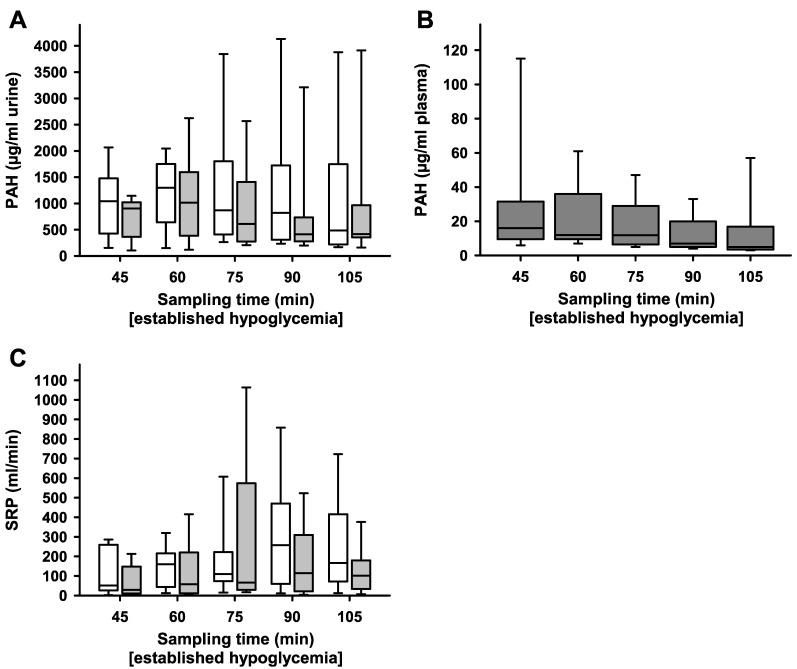
SRP and PAH after unilateral ablation of the renal nerves. Samples for PAH measurements were taken every 15 min during high volume infusion induced hyperperfusion of the kidneys after hypoglycaemia (blood glucose below 3 mmol/L) was established. (**A**) No differences in PAH [μg/mL] in urine during severe hypoglycaemia after unilateral ablation of renal nerves between renal-nerve-ablated kidneys (grey boxes) and non-ablated kidneys (white boxes); *n* = 9, *p* > 0.05. (**B**) Plasma PAH concentrations in samples drawn from the carotid artery during severe hypoglycaemia are well below the saturation concentration for the tubular secretion system; *n* = 9. (**C**) No differences in SRP [mL/min] during severe hypoglycaemia after unilateral ablation of renal nerves in non-ablated (white boxes) and ablated kidneys (grey boxes); *n* = 9, *p* > 0.05. Box plots show medians, 25/75th percentiles (boxes), and 10/90th percentiles (whiskers).

**Figure 4 ijms-24-12738-f004:**
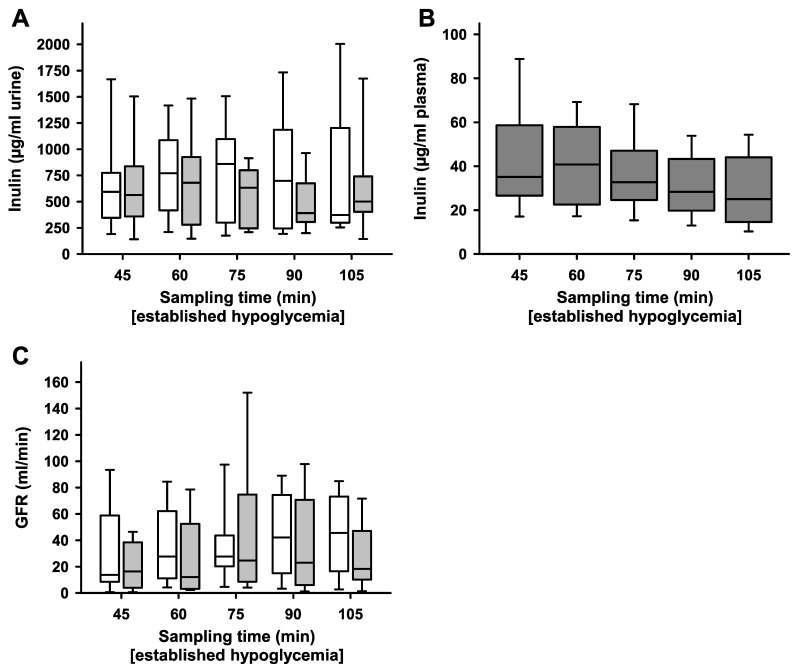
GFR and inulin after unilateral ablation of renal nerves. Each inulin sample and measurement of the glomerular filtration rate (GFR) were, respectively taken during hyperperfusion induced by high-volume infusion and during hypoglycaemia (blood glucose below 3 mmol/L) in 15 min intervals. (**A**) Inulin in urine samples [µg/mL] did not significantly differ between ablated (grey boxes) and non-ablated (white boxes) kidneys; *n* = 9, *p* > 0.05. (**B**) Inulin plasma samples drawn from the carotid artery; n = 9. (**C**) GFR [mL/min] did not differ significantly between non-ablated (white boxes) and ablated (grey boxes) kidneys; *n* = 9, *p* > 0.05. Box plots show medians, 25/75th percentiles (boxes), and 10/90th percentiles (whiskers).

**Figure 5 ijms-24-12738-f005:**
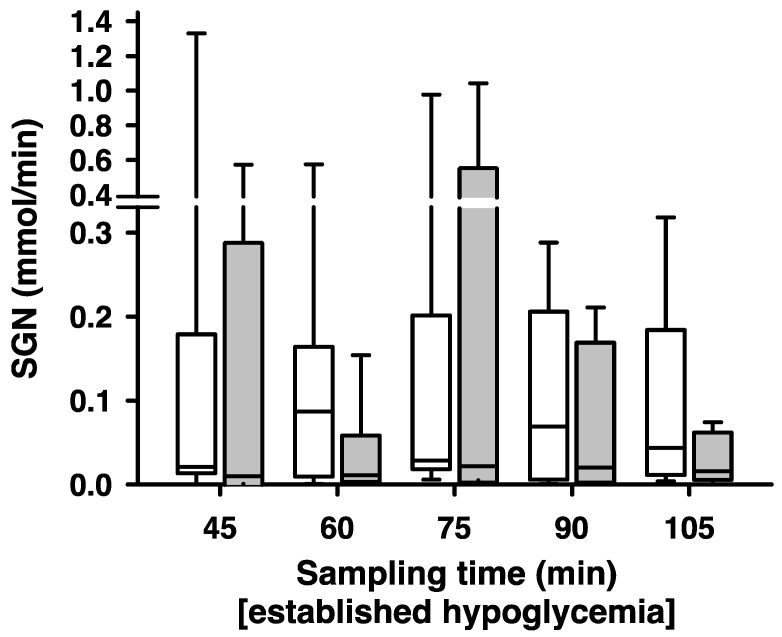
SGN after unilateral ablation of renal nerves. Each measurement of side-dependent renal net glucose release (SGN) taken, respectively in 15 min intervals during hyperperfusion induced by high-volume infusion and hypoglycaemia (blood glucose below 3 mmol/L). SGN [mmol/min] differed significantly between non-ablated (white boxes) and ablated (grey boxes) kidneys; *n* = 9, *p* = 0.004. Box plots show medians, 25/75th percentiles (boxes), and 10/90th percentiles (whiskers).

**Figure 6 ijms-24-12738-f006:**
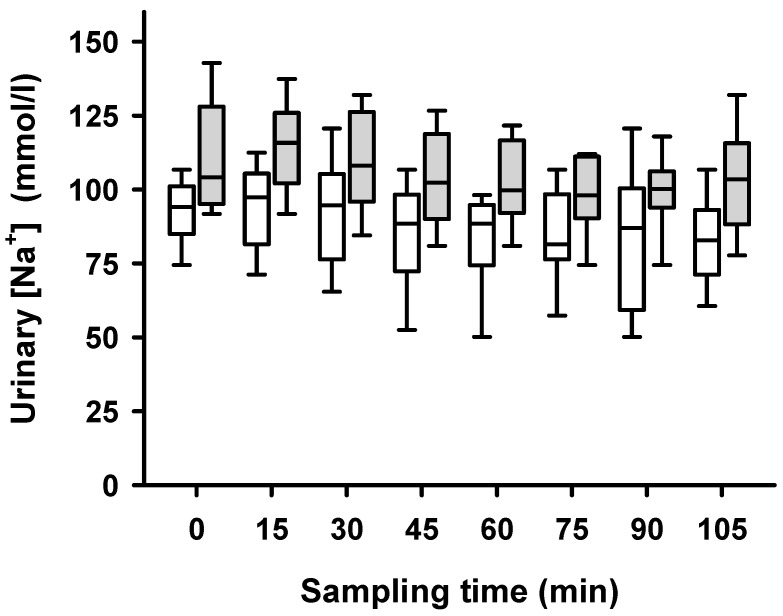
Urinary sodium concentrations after unilateral ablation of renal nerves. All samples of sodium excretion taken during hyperperfusion through high-volume infusion in 15 min intervals. Sodium concentrations of ablated (grey boxes) kidneys were significantly higher than that of non-ablated (white boxes) kidneys; linear mixed model; *p* = 0.019.

**Table 1 ijms-24-12738-t001:** Characteristics of laboratory animals and vital parameters.

Variables	Parameters
Sex (*n*_male_/*n*_female_)	0/9
Age (days)	78 ± 7
Weight (kg)	27 ± 6
Weight of ablated kidney (g)	104 ± 23
Weight of non-ablated kidney (g)	101 ± 22
Heart rate at baseline (bpm)	124 ± 7
Heart rate at the end of hypoglycemia (bpm)	212 ± 28
Blood pressure at baseline (systolic/diastolic) (mmHg)	108/70 ± 7/6
Blood pressure at the end of hypoglycemia (systolic/diastolic) (mmHg)	128/84 ± 7/3
Body temperature at baseline (°C)	37.6 ± 0.6
Body temperature at the end of hypoglycemia (°C)	35.8 ± 0.8
Data are given as means ± SD, *n* = 9	

## Data Availability

Raw data of side-dependent renal glucose release after unilateral denervation can be found in the Appendix A of this article.

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
