# Peer review of "Renal Glucose Release after Unilateral Renal Denervation during a Hypoglycemic Clamp in Pigs with an Altered Hypothalamic Pituitary Adrenal Axis after Late-Gestational Dexamethasone Injection"

_ijms, 2023, doi:10.3390/ijms241612738_

Round 1

Reviewer 1 Report

Dr. Nistor and colleagues provided a huge investigation of pathways underlying in body response to hypoglycemia. The authors applied high-level methods of fundamental physiological research. However, there are same issues, which must be solved, in order to publish this article.

Related to the design of the study:

1. The authors aimed to clarify how glucocorticoid exposure and nerve-ablation alter glucose release from the kidney. The experiment provided the comparison between nerve-ablated and non-ablated kidney. Then, the authors compared the present results with their previous study in discussion section. Nevertheless, at least statistical procedures must be performed in order to evidence significant of found changes. It’s also necessary to discuss if the experiment conditions, such as body weight, age, duration of pregnancy, body temperature were equal in this two studies.

2. The surgery was performed by complete unilateral removal of plexus renalis. However, authors did not note if they saved nerves to the ipsilateral adrenal gland unchanged. Denervation of adrenal gland could influence on the respond to hypoglycemia as well.

3. The authors noted the decrease of temperature during the surgery. However, the temperature at the surgery room and infusion solutions is not reported.

4. The authors claimed that difference between renal arterial and venous glucose reflects the gluconeogenesis in the kidney (Line 395-399). However, other factors, such as glucose consumption in renal cells and glucose reabsorbtion, can contribute to these changes. It is not clear how the authors distinguish this effects.

5. The authors investigated the changes in sodium excretion. But it is not clear how this part of the study follow to the aim of the study and how these findings supported the results.

6. The authors made a classical physiological acute experiment. However, there are not research of changes in glucose release and sodium reabsorbtion in kidney. The authors noted in Introduction section that changes in GLUTs and SGLTs can be involved in this process (Line 99). However, no experimental evidence was provided in this article.

7. The authors applied 9 pigs into the study. However, they presented the results as normally distributed and provided parametrical comparisons. The statistical procedures to check if the data distribution are normal must be perform.

Related to presentation of results:

1. The Authors used numbers to identify the time, which specimen of plasma or urine was collected. However, the figures contain different numbers of point. The notice of exact time of experiment (possibly 0 min, 15 min, 30 min…) can solve this problem.

2. The manuscript possibly contains typos and uncommon expressions, such as “p>0.01” (Line 33), “24.91±23.2” (Line 34, this the author rounded to 25% at Line 434), text from the template (Line 461-463).

Related to other parts of manuscript:

1. The Introduction looks quite excessive in description of relevance of the field, but did not provide the gaps in previous studies and how the current research can solve it.

2. The Conclusion contains some ideas, which were not investigated in the current research directly (Line 441-449).

Author Response

Dr. Nistor and colleagues provided a huge investigation of pathways underlying in body response to hypoglycemia. The authors applied high-level methods of fundamental physiological research. However, there are same issues, which must be solved, in order to publish this article.

Answer: We would like to thank the reviewer for the evaluation of the manuscript and for its kind reception.

Related to the design of the study:

  1. The authors aimed to clarify how glucocorticoid exposure and nerve-ablation alter glucose release from the kidney. The experiment provided the comparison between nerve-ablated and non-ablated kidney. Then, the authors compared the present results with their previous study in discussion section. Nevertheless, at least statistical procedures must be performed in order to evidence significant of found changes. It’s also necessary to discuss if the experiment conditions, such as body weight, age, duration of pregnancy, body temperature were equal in this two studies.

Answer:  We agree with the comments and have addressed them in the manuscript. We have performed further statistical analyses to clarify the mentioned differences of the respective glucose ratios in the discussion.  

In the following, we intend to address the experimental conditions of this manuscript and our prior study: The mother sow’s did not differ in respect to size, age and the food they received; similarly, there were no significant differences in length of gestation and piglet mortality ( Reference 25: Schiffner, R.; Rodriguez-Gonzalez, G.L.; Rakers, F.; Nistor, M.; Nathanielsz, P.W.; Daneva, T.; Schwab, M.; Lehmann, T.; Schmidt, M. Effects of Late Gestational Fetal Exposure to Dexamethasone Administration on the Postnatal Hypothalamus-Pituitary-Adrenal Axis Response to Hypoglycemia in Pigs. International journal of molecular sciences 2017, 18, doi:10.3390/ijms18112241.). Both the piglets utilised in our prior study investigating the effects of renal deneravation on renal glucose release without an altered HPA axis  (Reference 32: Bischoff, S.J.; Schmidt, M.; Lehmann, T.; Schwab, M.; Matziolis, G.; Saemann, A.; Schiffner, R. Renal glucose release during hypoglycemia is partly controlled by sympathetic nerves - a study in pigs with unilateral surgically denervated kidneys. Physiological reports 2015, 3, doi:10.14814/phy2.12603) and those of this current study were chosen from the pool of animals described in Reference 25.

Slight differences in the study population exist in the body weight of the piglets, which was 38±7kg in Reference 32 and is 27±6kg in this study, and the age at the time of the experiment, respectively 85±10 days vs. 78±7 days. The age difference can be assumed to be negligible due to the close proximity in respect to the developmental stage of the piglets. The difference in body can be explained in the context of the typical – and expected – changes effected through an alteration of the HPAA, and correspond to those observed in humans as well. Otherwise, the two sub populations did not exhibit any significant changes. The experimental protocol of Lit. 32 was adopted unaltered to a large extend in this current study. The body temperature of the animals which received a large surgical laparotomy access fell only moderately from von 37,5 ±0,5 to 36,1 ±0,7°C in Reference 32 and 37,6 ±0,6 °C to 35,8 ±0,8 °C  in this current study.

Action: We have added information pertaining to this in line 231-243 and line 384-386.

  1. The surgery was performed by complete unilateral removal of plexus renalis. However, authors did not note if they saved nerves to the ipsilateral adrenal gland unchanged. Denervation of adrenal gland could influence on the respond to hypoglycemia as well.

Answer: During the surgical preparation utilising an open abdominal access the neuroplexus of the arteria renalis was severed under visual control on the previously determined side. Fibres of the celiac ganglion which primarily innervate the adrenal gland were not manipulated A subsequent control of the success of the severing of the neuroplexus was not performed, however, the measured altered sodium excretion of the denervated kidney allows the conclusion that the renal denervation was successful in each individual case.

Action: Information were added to the discussion in line 420-426.

  1. The authors noted the decrease of temperature during the surgery. However, the temperature at the surgery room and infusion solutions is not reported.

Answer: The reviewer is correct to point out this oversight on our part.

Action: The relevant information was added to the manuscript in lines 394-45 and line 409.

  1. The authors claimed that difference between renal arterial and venous glucose reflects the gluconeogenesis in the kidney (Line 395-399). However, other factors, such as glucose consumption in renal cells and glucose reabsorbtion, can contribute to these changes. It is not clear how the authors distinguish this effects.

Answer: The reviewer is correct that our experimental setup does not allow to draw conclusions on the extent of glucose consumption of renal cells and glucose reabsorption. The experimental procedure consisted of a hypoglycemia below 3mmol/l achieved through insulin injections and a slow subsequent lowering of the blood glucose during the duration of the experiment. Furthermore, the experiment was preceded by a 24 hour food withdrawal. Taken together, these factors imply that the glucose consumption of renal cells and reabsorption appear negligible in our experimental setting, since our calculations and comparisons with the non-denervated organs comprise summations that include all of the aforementioned factors (which should not significantly differ between denervated and non-denervated organ). Insofar, our experimental setup is not capable to elucidate questions regarding local glucose consumption of renal cells or side-dependent glucose reabsorption.

Action: We have added a clarifying statement pertaining to the above in lines 269-274.

  1. The authors investigated the changes in sodium excretion. But it is not clear how this part of the study follow to the aim of the study and how these findings supported the results.

Answer: Renal sodium excretion is a well-established physiological parameter to determine the success of a denervation procedure of a kidney, due to the circumstance that an increased sodium excretion causes a concomitantly elevated water excretion, followed by a decrease of blood pressure. We chose this parameter as a functional variable that allowed for a conclusion on the success of the denervation procedure.

Action: We have clarified this matter in lines 312-316.

  1. The authors made a classical physiological acute experiment. However, there are not research of changes in glucose release and sodium reabsorbtion in kidney. The authors noted in Introduction section that changes in GLUTs and SGLTs can be involved in this process (Line 99). However, no experimental evidence was provided in this article.

Answer: The reviewer is correct that our experimental setup does not allow to draw conclusions on conceivable changes of receptor density effected by the prenatal glucocorticoid inection and long-term alteration of the HPAA. However, elucidating these mechanisms was not the purpose of this particular investigation. The reference to GLUTs and SGLTs in the introduction was intended to provide the reader with a general overview of the physiological mechanisms involved in renal glucose homeostasis.

  1. The authors applied 9 pigs into the study. However, they presented the results as normally distributed and provided parametrical comparisons. The statistical procedures to check if the data distribution are normal must be perform.

Answer: We acknowledge the reviewer's concern regarding the sample size of 9 animals, which may not be sufficient to assume the central limit theorem and normality of the averaged data. In response to this valid concern, we performed a Shapiro-Wilk test to assess the normality of the data. The results indicated that the data in the non-ablation group was normally distributed, but the ablation group were non-normally distributed.

As a result of these findings, we opted to use a more robust statistical approach, namely the Wilcoxon Signed-Rank Test, which does not rely on the assumption of normality. This test allowed us to properly analyze the data and compare the SGN values between ablated and non-ablated animals.

Interestingly, the change in statistical approach did not alter the result significantly, as we still observed a significant difference between the SGN of ablated and non-ablated animals (p=0.004).

We express our sincere gratitude to the reviewer for raising this crucial point, leading us to conduct an improved statistical analysis. Throughout the manuscript, we have made the necessary corrections to accurately report the results and describe the revised statistical method accordingly.

Action: We have changed and expanded the “Statistics” section in “Methods” accordingly in lines 492-512.

Related to presentation of results:

  1. The Authors used numbers to identify the time, which specimen of plasma or urine was collected. However, the figures contain different numbers of point. The notice of exact time of experiment (possibly 0 min, 15 min, 30 min…) can solve this problem.

Answer: The reviewer is correct that the numbering of time stamps in our figures was confusing to an extent.

Action: We have changed the captions of the figures to achieve greater clarity.

  1. The manuscript possibly contains typos and uncommon expressions, such as “p>0.01” (Line 33), “24.91±23.2” (Line 34, this the author rounded to 25% at Line 434), text from the template (Line 461-463).

Answer:  We thank the reviewer for bringing the identified error to our attention. Indeed, the correct statement in the abstract should be p<0.05 instead of p>0.01. We deeply regret the oversight that led to this mistake and apologize for any confusion it may have caused. 

Action: We have changed the respective passages in line 33 and line 462 and corrected a number of small typos throughout the manuscript.

Related to other parts of manuscript:

  1. The Introduction looks quite excessive in description of relevance of the field, but did not provide the gaps in previous studies and how the current research can solve it.

Answer: We understand the reviewer’s sentiment as to the length of the introduction. However, as the reviewer acknowledged themself in the introductory remark to the review, the study covers a relatively large topic that combines a number of matters from the body’s stress systems – HPAA and ANS – to the procedure under investigation – renal denervation – and the specific clinical context and its potential implications on a certain patient group – respectively hypertension, and the diabetic patient, specifically those in danger of recurrent hypoglycemic episodes. To ensure a comprehensive introduction and discussion of all the necessary relevant components of the research question, we deemed it necessary to extend the introduction to its current length.

Action: We have clarified the gaps in the current state of research – otherwise implied within the text – specifically in lines 64-69 and lines 118-122.

The Conclusion contains some ideas, which were not investigated in the current research directly (Line 441-449).

Answer: In the conclusion, we attempted to give an outlook on the potential translational aspects should our findings be confirmed and broadened in subsequent studies. However, we would posit that those potential implications are adequately phrased in a hypothetical mode (“could” etc.) as not to confuse the reader or to imply the presence of demonstrative results of these deliberations within the study.  

Reviewer 2 Report

The authors investigated the influence of an altered HPAA on renal glucose release during hypoglycemia. They primarily assessed side-dependent renal net glucose release (SGN) during hypoglycemia in prenatal dexamethasone-induced HPAA pigs.They demonstrated that  SGN decreased in ablated kidney and it was only half of that observed in non-HPAA-altered animals in their previous investigation.

The findings are interesting, however the reviewer raise some points as below.

Clinical Significance:

The findings are interesting, however it is difficult to image how it is applicated to a clinical setting. If caution is necessary during renal denervation in DM patients, conducting the study using DM model animals would likely yield clearer results. Please provide a clear description of the specific patient n and clinical context that the authors are targeting.

Control Group:

In the comparison with the authors' previous work, were various study conditions including age, sex, and others similar to in the current study?

Statistics:

In Figure 5, SGN appears to have a large standard deviation (SD). Please specify the large SD.

Additionally, the 60-minute values seem statistically insignificant.

What statistical analysis was performed?

Please present the raw data of SGN for all nine subjects at each time point in a supplemental table.

Renal Denervation:

Could you clarify which kidney (left or right) underwent denervation?

ACTH and COR:

Can you provide data on ACTH and COR levels at baseline and during hypoglycemia?

Other Factors Involved in Gluconeogenesis:

Please discuss factors related to gluconeogenesis other than renal blood flow and HPAA.

In particular, the effects of dexamethasone administration on the liver.

Author Response

The authors investigated the influence of an altered HPAA on renal glucose release during hypoglycemia. They primarily assessed side-dependent renal net glucose release (SGN) during hypoglycemia in prenatal dexamethasone-induced HPAA pigs.They demonstrated that  SGN decreased in ablated kidney and it was only half of that observed in non-HPAA-altered animals in their previous investigation.

 The findings are interesting, however the reviewer raise some points as below.

 Answer: We would like to thank the reviewer for the evaluation of the manuscript and for its kind reception.

Clinical Significance:

The findings are interesting, however it is difficult to image how it is applicated to a clinical setting. If caution is necessary during renal denervation in DM patients, conducting the study using DM model animals would likely yield clearer results. Please provide a clear description of the specific patient n and clinical context that the authors are targeting.

Answer: We understand the reviers objection. However, this current study aimed to investigate the fundamental consequences of prenatal glucocorticoid treatments on later-in-life renal gluconeogenic response to (hypoglycemic) stressors, it can therefore be seen as elemental research. We concur, nonetheless, that future research should be conducted in disease models specifically adapted to transferability of the rendered results and the pathophysiological correlate in humans. As for the limitations of the study and a description of the human patient affected by these mechanisms (and the conditions under which they might occur), we consider the relevant paragraphs already in the manuscript to be comprehensive and clear in that regard (lines 277-344).

Action: We have added a sentence stressing the importance of adapted experimental models in further research into the matter under investigation in lines 372-376.

Control Group:

In the comparison with the authors' previous work, were various study conditions including age, sex, and others similar to in the current study?

Answer: We agree with the researcher’s objection that the previous experimental conditions were, with the mere reference to our prior study, not accessible enough in the text.

Action: We have added the required information in lines 384-386 and 231-243.

Statistics:

In Figure 5, SGN appears to have a large standard deviation (SD). Please specify the large SD.

Additionally, the 60-minute values seem statistically insignificant.

Answer: Due to the low number of animals, we have not tested for differences in renal gluconeogenesis for individual time points. Instead, we combined the different measurements from the different time points into a mean value per animal per side and then compared this between groups using a Wilcoxon Signed-Rank Test.

Action: We have expanded the description of the methods to make this point clearer (see below).

What statistical analysis was performed?

Please present the raw data of SGN for all nine subjects at each time point in a supplemental table.

Answer: Information regarding the utilised statistical model have been expanded upon by our biostatistician in the “Statistics” section of “Methods”. Furthermore, we have added a table 2 as supplementary material. Regarding the data, the reviewer is quite correct in pointing out the large spread of results, which is due to the great inter-individual breadth of gluconeogenic response and our decision to leave the two greatly heightened values (potential technical errors) in the statistical analysis (please see General comment to both reviewers).

Action: We have changed and expanded the “Statistics” section in “Methods” accordingly in lines 492-512, and now provide the raw data in a table 2 as supplementary material.

Renal Denervation:

Could you clarify which kidney (left or right) underwent denervation?

Answer: The side of renal denervation was determined by a randomisation procedure to prevent bias. Information (and its discussed consequences as a limiting factor) regarding this can be found in lines 291-296 of the manuscript, as well as in the now added supplementary

ACTH and COR:

Can you provide data on ACTH and COR levels at baseline and during hypoglycemia?

Answer: In this study we utilised animals from a pool of offspring from which we furthermore analysed a subgroup in a prior experiment in regard to ACTH and cortisol levels both before and during a hypoglycemic episode (albeit without a renal denervation procedure. In this experimental setup, we did not conduct a further analysis of the hormonal response. The reviewer is correct to point this out as a limitation of this study. However, since we did not manipulate the nerval fibres of the celiac ganglion or operated on the adrenal glands, we did not expect a significant change in hormonal response solely from performed surgical procedure. 

Action: We have clarified this matter and provide an explanation in lines 227-231.

General note two both reviewers:

In updating and clarifying Figure 5 (SGN), we unfortunately noticed a graphical error in the figure through a fault of our own. We have corrected this unfortunate oversight, which we didn’t notice before our initial submission, and apologise for the confusion. The underlying statistical data on which the graphical presentation has been, and is based, have remained valid and unchanged. We have added a clarification and discussion of the visually quite apparent outliers at two consecutive time points (specifically noticed beforehand by reviewer two). These two statistical outliers are due to two experimental animals which, at two consecutive measurements, exhibited an (for us not easily explainable) increased side-dependent plasma flow, which lead to a concomitant increase in GFR, urine volume, and by extension, due to the utilised method of calculation, also side-dependent renal glucose release. We discuss this issue in lines 244-248 of the manuscript.

Other Factors Involved in Gluconeogenesis:

Please discuss factors related to gluconeogenesis other than renal blood flow and HPAA.

In particular, the effects of dexamethasone administration on the liver.

Answer: Please find a discussion of other factors involved in renal gluconeogenesis in lines 103-107. In regard to the effects of glucocorticoids and dexamethasone specifically we have added a general reference to potential underlying factors in lines 280-286. However, our experimental setup was not designed to account for potential hepatic mechanisms or to elucidate hepatorenal reciprocity, one supposed underlying cause of the mitigated renal response observed in our study. A more expansive discussion of the intricacies of the hepatic glucose metabolism would therefore be somewhat unrelated to the actual experimental intervention and the primary subject of this manuscript.

Action: We have added a general reference to the potentially underlying factors in lines 289-296.

Round 2

Reviewer 1 Report

The authors modified the munuscript according to reviewer's notes. Nevertheless, some minor changes can be made:

1. The authors found that some raw data are not distrisbuted normally. Nevertheless, the revised virsion contains data presented as normally distributed (means±SEM). However, presentation of non-normally distributed data as medias and range could be more sutable. The rivised virsion also does not contain information about appyed Shapiro-Wilk test.

2. According to the last reviewer remark: the scientific editor can judge if the current conclusion (Line 527-532 in the revised virsion) has been supported by the current research and sutable for research paper.

Author Response

  1. The authors found that some raw data are not distrisbuted normally. Nevertheless, the revised virsion contains data presented as normally distributed (means±SEM). However, presentation of non-normally distributed data as medias and range could be more sutable. The rivised virsion also does not contain information about appyed Shapiro-Wilk test.

Answer:

The reviewer is correct that the statistical analysis, which confirmed that the results were partly not normally distributed, was formerly not specifically referenced in the Methods part of the manuscript.

Furthermore, we agree that the presentation as medias and range or more suitable.

Action:

We have added information regarding the application of the Shapiro-Wilk to lines 494-495. Additionally, we have altered the figures representing data that are not normally distributed to now depict medians and range as boxplots and have altered the figure legends accordingly.

  1. According to the last reviewer remark: the scientific editor can judge if the current conclusion (Line 527-532 in the revised virsion) has been supported by the current research and sutable for research paper.

Answer:

We would concur to refer that specific decision to the editor, as we still, respectfully, believe that the statement is formulated in a suitably hypothetical tone that does not imply a factual basis supported by experimental/statistical results.

On a general note: We have noticed a minor inconsistency after the previous review round in respect to the Supplementary material and the new, specifications of experimental time points in the figures. We have altered the temporal signifiers of the raw data in the supplementary material to correspond with those of the figures (in mins after the start of the experimental procedure rather than as abstract measuring points).

We would like to thank the reviewer for their constructive criticism, helpful recommendations and general advice, which, undoubtedly, increased the overall quality of the manuscript.

Reviewer 2 Report

The reviewer found that the authors responded sincerely to the comments in a short period.

Their responses are sincere and accurate.

Thus, the reviewer considers that the revised manuscript is acceptable for publication.

There are no further comments.

Congratulations.

Author Response

The reviewer found that the authors responded sincerely to the comments in a short period.

Their responses are sincere and accurate.

Thus, the reviewer considers that the revised manuscript is acceptable for publication.

There are no further comments.

Congratulations.

Answer:

We would like to thank the reviewer for their constructive criticism, helpful recommendations and general advice, which, undoubtedly, increased the overall quality of the manuscript.